# Implementation of a Conditional Latent Diffusion-Based Generative Model to Synthetically Create Unlabeled Histopathological Images

**DOI:** 10.3390/bioengineering12070764

**Published:** 2025-07-15

**Authors:** Mahfujul Islam Rumman, Naoaki Ono, Kenoki Ohuchida, Ahmad Kamal Nasution, Muhammad Alqaaf, Md. Altaf-Ul-Amin, Shigehiko Kanaya

**Affiliations:** 1Computational Systems Biology Laboratory, Division of Information Science, Nara Institute of Science and Technology, Nara 630-0192, Japan; kamal.nasution@naist.ac.jp (A.K.N.); muhammad.alqaaf_subandoko.mb5@is.naist.jp (M.A.); amin-m@is.naist.jp (M.A.-U.-A.); skanaya@gtc.naist.jp (S.K.); 2Data Science Center, Nara Institute of Science and Technology, Nara 630-0192, Japan; 3Department of Surgery and Oncology, Kyushu University, Fukuoka 819-0395, Japan; ouchida.kenoki.060@m.kyushu-u.ac.jp

**Keywords:** diffusion models, encoder-decoder architectures, histopathology image processing, image generation, artificial intelligence, deep learning

## Abstract

Generative image models have revolutionized artificial intelligence by enabling the synthesis of high-quality, realistic images. These models utilize deep learning techniques to learn complex data distributions and generate novel images that closely resemble the training dataset. Recent advancements, particularly in diffusion models, have led to remarkable improvements in image fidelity, diversity, and controllability. In this work, we investigate the application of a conditional latent diffusion model in the healthcare domain. Specifically, we trained a latent diffusion model using unlabeled histopathology images. Initially, these images were embedded into a lower-dimensional latent space using a Vector Quantized Generative Adversarial Network (VQ-GAN). Subsequently, a diffusion process was applied within this latent space, and clustering was performed on the resulting latent features. The clustering results were then used as a conditioning mechanism for the diffusion model, enabling conditional image generation. Finally, we determined the optimal number of clusters using cluster validation metrics and assessed the quality of the synthetic images through quantitative methods. To enhance the interpretability of the synthetic image generation process, expert input was incorporated into the cluster assignments.

## 1. Introduction

Generative artificial intelligence (GAI) has emerged as one of the most transformative technological advancements in recent years. From text generation and image synthesis to code automation and molecular design, generative models are reshaping industries, redefining creativity, and challenging conventional thinking. In particular, generative AI has revolutionized image synthesis, opening new frontiers in creativity, automation, and practical applications. Leading this progress are diffusion models, a class of generative models that have demonstrated remarkable abilities in producing high-fidelity images, inpainting, super-resolution, and conditional synthesis. By employing iterative denoising processes inspired by thermodynamics, diffusion models have surpassed traditional generative adversarial networks (GANs) and variational autoencoders (VAEs) in terms of sample quality, stability, and controllability.

The remarkable capability of diffusion models to generate high-quality images has been highlighted in numerous recent research articles. Ho et al. first proposed denoising diffusion probabilistic models (DDPMs) as a powerful alternative to GANs for high-quality image synthesis [1]. Unlike GANs, which often suffer from mode collapse and training instability [2], diffusion models offer more predictable training, making them more robust and reliable for large-scale image generation tasks. Rombach et al. later proposed latent diffusion models (LDMs) [3]. Unlike standard diffusion models that operate directly in pixel space, LDMs compress images into a lower-dimensional latent space before applying the diffusion process, significantly reducing computational complexity while maintaining high-quality generative outputs. The authors of [4] presented techniques to improve the performance of DDPMs in terms of efficiency, sampling speed, and likelihood estimation, all while preserving high-quality generative performance. Furthermore, the findings in [5] suggest that DDPMs can automatically and optimally adapt to unknown low-dimensional structures within data, thereby enhancing their practical efficiency.

In addition to the impressive capabilities offered by GAI models, ensuring the integrity and reliability of generated or derived features is a critical concern, especially in sensitive domains such as medical imaging. Recent studies have shown that even minor corruption, such as noisy, false, or mislabeled features, can significantly degrade the performance of both classical and deep learning classifiers, regardless of the model architecture [6]. This vulnerability underscores the importance of robust feature engineering and the careful validation of synthetic data pipelines. In the context of histopathological image analysis, where ground truth annotations are often scarce, the use of unsupervised or synthetic representations requires particular attention to data quality, as corrupted latent features can silently undermine downstream classification and interpretation. These concerns further motivate the development and evaluation of conditional generative models that not only synthesize realistic images but preserve structural fidelity and robust feature representations.

Another study has highlighted several critical vulnerabilities associated with the use of synthetic medical data generated by artificial intelligence [7]. AI models trained on such data can be highly sensitive to its quality and representativeness, which may hinder their ability to generalize to real-world clinical scenarios. Furthermore, the process of generating synthetic data may inadvertently omit rare but important clinical features, perpetuate existing biases, or introduce artificial patterns that undermine model reliability. In addition, models trained on synthetic datasets may produce explanations that differ from those based on real data, thereby challenging interpretability and clinical trust. These vulnerabilities underscore the need for the rigorous evaluation and validation of generative AI systems intended for medical applications.

While it is important to recognize the potential vulnerabilities associated with GAI, the adoption of LDMs offers several distinct advantages that help address these concerns. LDMs operate in a compressed latent space, which not only enhances computational efficiency but enables the generation of high-fidelity, structurally realistic synthetic images. This latent-space approach helps preserve both global and local features, making the synthetic data more representative of true biological variability. Moreover, conditional LDMs allow for flexible and controllable synthesis without the need for explicit annotations, enabling the creation of new data that reflects real-world complexity. By employing robust perceptual and structural similarity metrics during training, both the realism and feasibility of generated images can be ensured. Consequently, LDMs represent a promising direction for generating high-quality, privacy-preserving, and application-specific synthetic medical data, which can improve model training, validation, and interpretability in challenging domains such as histopathology.

In our research, we implemented a conditional latent diffusion model (cLDM) to generate synthetic histopathology images. First, we trained a VQ-GAN to compress high-dimensional image data into a lower-dimensional latent space. Next, we performed a diffusion process on the pre-trained latent space, in which noise was added to the latent features and a reverse process was learned to reconstruct clean latent representations over multiple steps. We then applied clustering to these latent features and used the clustering results to guide the diffusion model in generating specific outputs. This conditioning approach was necessary due to the absence of labels in our dataset. Subsequently, we determined the optimal number of clusters using cluster validation metrics. Finally, we generated a large number of synthetic images in the context of each cluster set and evaluated the quality of the generated images using several metrics to assess their similarity to the original references.

## 2. Related Work

In this section, we discuss several recent works that have utilized diffusion models. In [8], the authors presented a novel application of DDPMs in wireless communications, demonstrating their potential to enhance system robustness and signal reconstruction quality in the presence of hardware-induced distortions and challenging transmission environments. Although the study focused on specific hardware impairments, further research may be needed to evaluate the model’s effectiveness across a wider range of real-world scenarios or heterogeneous hardware setups. Podell et al. implemented a U-Net architecture three times larger than its predecessors in a text-to-image latent diffusion model, resulting in a framework for high-fidelity text-to-image synthesis [9]. This work introduced effective architectural and training improvements to latent diffusion models, significantly enhancing high-resolution image synthesis quality. However, these enhancements may increase computational requirements and necessitate further validation on diverse, real-world datasets. The authors of [10] addressed the challenge of controllable image synthesis by using user-provided scribbles along with text prompts to produce high-fidelity, semantically rich images, overcoming the limitations in previous guided image synthesis approaches within the latent diffusion framework. While the robust guidance mechanism of this model enables high-fidelity and controllable image synthesis, it may require extensive tuning, and its performance on out-of-domain or low-resource scenarios has not been fully explored. Another work demonstrated that the selective fine-tuning of diffusion models can produce high-quality images under differential privacy constraints, indicating a promising direction for privacy-preserving generative models [11]. While differential privacy enables image generation without significant quality loss, it may still introduce trade-offs in model utility.

Zhang et al. investigated the resilience of LDMs against adversarial attacks [12]. Their work provides a comprehensive analysis of the robustness of latent diffusion models under various perturbations, offering valuable insights into their reliability. However, the study primarily focuses on synthetic benchmarks, so its findings may not fully reflect the challenges encountered in real-world or domain-specific applications. The authors of [13] proposed Smooth Diffusion, a novel approach that not only improves text-to-image generation but enhances performance across several downstream tasks, including image interpolation, inversion, and editing. Nonetheless, the proposed methods may introduce additional computational overhead, and their effectiveness across diverse data modalities requires further validation. In [14], the contributors presented a new application of latent diffusion models in geological modeling, providing a robust framework for efficient parameterization and data assimilation in facies-based geomodels. However, the scalability of this approach to more complex geological settings and its integration with real field data remain to be fully established. Cola-Diff, a conditional latent diffusion model, introduces a unique network architecture designed to reduce compression artifacts and noise in the latent space while processing multiple MRI inputs [15]. However, the model’s generalization to rare pathologies and its robustness across diverse clinical datasets require further evaluation.

Yuan et al. investigated the use of conditional latent diffusion models (CLDMs) for image restoration (IR) tasks [16]. Their work rigorously evaluates the effectiveness of CLDMs across various image restoration applications, demonstrating strong performance compared to traditional methods. However, the study also notes limitations in handling highly degraded images, and identifies cases where restoration quality may still fall short of specialized techniques. In [17], the authors leveraged conditional neural fields and latent diffusion models to capture the complex dynamics of turbulence, enabling the generation of realistic flow patterns with both computational efficiency and accuracy. This methodology shows significant promise for advancing simulations in fluid dynamics and related fields, though its applicability to turbulence scenarios beyond those seen during training and its computational requirements warrant further investigation. Another innovative approach, introduced by the researchers in [18], enhances the diversity of images generated from textual descriptions by integrating autoregressive latent priors into diffusion models. Specifically, Kaleido diffusion improves conditional diffusion models through autoregressive latent modeling, resulting in more coherent and detailed image synthesis. Nevertheless, the increased complexity may lead to longer inference times, and its adaptability to tasks beyond standard image synthesis remains to be demonstrated.

The authors of [19] utilized LDMs for the conditional generation of drug-like small molecules with optimized properties. Their approach addresses the complex multi-parameter optimization challenges inherent in therapeutic design, such as balancing potency, selectivity, bioavailability, and safety. However, the real-world applicability and experimental validation of the generated molecules’ properties remain limited and require further investigation. Zhuang et al. introduced a method that extends diffusion probabilistic models to learn distributions over continuous functions defined on metric spaces, commonly referred to as fields [20]. Their work presents a unified framework capable of handling diverse data types, including 2D images, 3D geometries, and fields on non-Euclidean spaces, without relying on latent vector representations. Nonetheless, the increased model complexity may pose scalability challenges, and comprehensive real-world evaluations are still needed. In [21], the authors explored the application of diffusion probabilistic models to synthesize high-quality 3D medical imaging data. Their results indicate that the diffusion model effectively captures complex anatomical details, producing images that closely resemble real medical scans. However, challenges remain in scaling these models for large clinical datasets and ensuring the preservation of rare or subtle pathological features. Dar et al. evaluated the potential of 3D LDMs on two medical imaging datasets and found that the LDMs can memorize and replicate training data in situations with limited data availability [22]. This finding highlights the need for improved privacy-preserving techniques. Another study found that, while LDMs tend to memorize patient data, they generally outperform non-diffusion models in synthesis quality [23], emphasizing the current limitations of existing mitigation strategies.

The researchers in [24] employed diffusion models to generate synthetic medical images for training convolutional neural networks (CNNs) in medical image analysis tasks. Their work addresses challenges such as privacy concerns and data scarcity, while also highlighting recent advancements and diverse applications in the field. In [25], the authors introduced Med-cDiff, a conditional DDPM designed for medical image generation tasks, such as super-resolution, denoising, and inpainting, addressing the limitations in existing generative models, including training stability in GANs and blurriness in VAE-generated images. This model achieves high-quality and diverse outputs across multiple modalities; however, its reliance on condition labels may limit its applicability in fully unsupervised scenarios. In another study, the authors proposed a technique using latent diffusion models to generate normal counterparts of abnormal medical images, thereby enabling the identification of diagnostically relevant components by comparing the original and generated images [26]. Nevertheless, the effectiveness of this approach across diverse imaging modalities and its integration with existing clinical workflows requires further investigation. In [27], the researchers evaluated the effectiveness of LDMs across datasets, such as breast MRI, hand X-rays, head CT scans, and chest X-rays, aiming to produce high-quality synthetic images that can augment existing medical datasets. However, this approach still needs further evaluation regarding the preservation of rare clinical features and its impact on downstream diagnostic performance.

Pinaya et al. investigated the use of LDMs to generate high-resolution 3D synthetic brain images [28]. Their study demonstrates the ability of LDMs to produce high-quality and anatomically plausible brain images. However, further research is needed to assess their robustness to pathological variations and generalizability across different imaging protocols. The authors of [29] introduced the Medfusion framework, a latent DDPM aimed at enhancing medical image synthesis as an alternative to traditional GAN methods. While this work offers valuable insights, it also reveals modality-specific limitations, and suggests that no single model universally outperforms the other across all medical imaging tasks. Han et al. proposed ClusDiff, a novel framework designed to generate high-quality synthetic food images to facilitate more accurate and efficient image-based dietary assessments [30]. Nonetheless, the method’s reliance on clustering quality may hinder performance in datasets with ambiguous or overlapping food categories. In [31], a diffusion probabilistic model was implemented to generate high-quality synthetic histopathology images, with a particular focus on brain cancer tissues. However, its generalizability to diverse staining methods and rare tissue patterns remains to be thoroughly evaluated. Another study presented a method for creating synthetic histopathological whole slide images using diffusion models [32]. This work highlights the capability of diffusion-based models to generate high-quality, gigapixel-scale histopathological whole slide images, advancing the field of digital pathology. Nevertheless, computational intensity and validation on diverse histopathological subtypes and clinical scenarios remain significant challenges. The authors of [33] introduced PathLDM, a novel LDM designed to generate high-quality histopathology images conditioned on textual descriptions. This model enables controllable and semantically meaningful image generation from textual prompts; however, its reliance on high-quality text annotations may limit its applicability in datasets lacking comprehensive or standardized labeling.

Beyond diffusion models, deep learning has profoundly transformed the field of image processing, driving significant advances in generative modeling, representation learning, and real-world applications across diverse domains. Early breakthroughs include the development of the VAE by Kingma and Welling [34], which established a probabilistic framework for learning latent representations and generating novel images. VAEs provide a strong foundation for unsupervised representation learning, enabling downstream tasks such as image synthesis and anomaly detection. The introduction of GANs and their numerous variants further revolutionized image synthesis and translation. Notably, the StyleGAN architecture proposed in [35] utilized style-based modulation to achieve unprecedented image quality and controllability, particularly for high-resolution images. Meanwhile, the pix2pix framework in [36] leveraged conditional GANs for image-to-image translation, providing a unified approach for tasks such as semantic segmentation, colorization, and domain adaptation. Collectively, these generative models have become indispensable for producing realistic and diverse images across a wide range of applications.

Beyond generative modeling, deep learning architectures have greatly advanced the field of image segmentation. The U-Net model introduced by Ronneberger et al. [37] has become the de facto standard for biomedical image segmentation due to its encoder–decoder structure and extensive use of skip connections to preserve spatial information. This architecture was further enhanced by the DeepLab family of models [38], which introduced atrous convolution and conditional random fields (CRFs) to refine semantic segmentation, particularly in complex scenes. The emergence of self-supervised and unsupervised learning paradigms has also played a crucial role in improving visual representation learning. Methods such as SimCLR [39] and MoCo [40] exemplify the effectiveness of contrastive learning, enabling neural networks to learn rich and transferable features from unlabeled data through augmentation-based positive and negative sample mining. These approaches have demonstrated strong performances across various image classification and retrieval tasks. Transformer architectures, previously dominant in natural language processing, have also made significant inroads into computer vision. The Vision Transformer (ViT) proposed by Dosovitskiy et al. [41] applied the self-attention mechanism to image patches, allowing the model to capture long-range dependencies and global context more effectively than traditional CNNs. The ViT and its derivatives have achieved state-of-the-art results in image classification, inspiring a new wave of transformer-based vision models.

Within the medical domain, deep learning solutions have addressed the challenges posed by limited or imperfect datasets. The authors of [42] provided a comprehensive review of deep learning approaches tailored to medical image segmentation, highlighting strategies for mitigating data quality issues and improving robustness. Significant progress has also been made in object detection, with models such as YOLOv3 [43] offering real-time, accurate detection suitable for both general and specialized image analysis tasks. Collectively, these advancements have established a strong foundation for modern image processing, pushing the boundaries of what is possible with deep learning. Progress has spanned generative modeling, image-to-image translation, segmentation, unsupervised and self-supervised learning, and the development of efficient architectures for real-world deployment. As the field continues to evolve, the integration of these methodologies promises even greater impact across a wide range of domains.

## 3. Materials and Methods

### 3.1. Description of the Dataset

In this study, we used histopathology images of KPC mice as our research dataset. The Department of Surgery and Oncology at the Graduate School of Medical Sciences, Kyushu University, provided these images. The dataset is unlabeled and consists of images stained using the Hematoxylin and Eosin (HE) technique, which is the most commonly used staining method in histopathology for detailed visualization of cell structures and tissues. Hematoxylin stains the cell nuclei purple, while eosin stains the fibers pink [44]. The dataset contains a total of 11,000 images, of which 10,000 were used for training the model and 1000 were reserved for testing. All images have a resolution of 128 × 128 pixels.

### 3.2. Vector Quantized Generative Adversarial Network (VQ-GAN)

We selected the VQ-GAN as the autoencoder framework for our latent diffusion model. Using an autoencoder in the LDM makes the diffusion process more efficient, as it is applied in the image’s latent space rather than in the pixel space, thereby reducing computational requirements. The VQ-GAN is a generative model that combines vector quantization (VQ), autoencoders, and GANs to efficiently generate high-quality images while operating in a discrete latent space. It is an improved version of the VQ-VAE (Vector Quantized Variational Autoencoder), incorporating adversarial training to produce sharper and more realistic images. The inclusion of a discriminator and a discretized latent space in the VQ-GAN results in less blurry images compared to conventional VAEs [45].

Our VQ-GAN model consists of three main components: the VQ-VAE architecture, the LPIPS (learned perceptual image patch similarity) model, and the PatchGAN discriminator. First, we briefly describe the VQ-VAE network. In [46], the VQ-VAE model was introduced, pioneering the use of a discrete codebook to represent latent variables. Unlike traditional VAEs, which use continuous latent variables, the VQ-VAE quantizes latent representations using a learned codebook of discrete embeddings. This approach makes the autoencoder well-suited for applications such as image generation, speech synthesis, and reinforcement learning. The VQ-VAE loss function consists of three components as follows:(1)$$\;L_{VQ-VAE}=L_{rec}+L_{cb}+\beta L_{commit}$$

Here, $L_{rec}$ is the reconstruction loss, measured as the mean squared error (MSE) between the original image and the reconstructed image; $L_{cb}$, or codebook loss, updates the codebook embeddings by bringing them closer to the encoded latent representations; $L_{commit}$ is the commitment loss, which encourages the encoder’s latent outputs to remain close to the discrete codebook vectors and prevents codebook collapse. The parameter β is the commitment coefficient, with values typically chosen between 0 and 1. For our experiment, we selected 0.2 as the value of β.

Secondly, an LPIPS model is used to measure the perceptual loss in the VQ-GAN framework. Perceptual loss ensures that the generated images not only match the pixel values of the original images but preserve high-level features, such as textures and structure. By comparing deep feature representations rather than raw pixel values, perceptual loss helps produce sharper and more realistic images. To calculate this loss, high-level feature maps of both the original and the generated images are extracted from a pre-trained neural network (such as VGG or ResNet), and the perceptual distance between them is computed—a concept introduced in [47]. In our work, we used a pre-trained VGG-16 model via the PyTorch (version 2.4.1) deep learning library [48]. Feature representations were extracted at various layers of the VGG-16, and the L2 distance was computed between the corresponding features. We then applied normalization and linear weighting layers before aggregating the differences. Finally, the perceptual loss was averaged over the spatial dimensions.

Finally, we describe the discriminator used in our VQ-GAN model. The discriminator plays a crucial role in enhancing the realism and sharpness of generated images by enforcing adversarial training [49]. In adversarial training, two neural networks, a generator and a discriminator, are trained in a competitive setting. The objective is to improve the quality of the generated outputs by encouraging the generator to produce increasingly realistic samples. In our experiment, we used a PatchGAN discriminator, while the decoder of the VQ-VAE serves as the generator. Unlike traditional discriminators that classify an entire image as real or fake, the PatchGAN discriminator divides the image into smaller patches and classifies each patch independently [50]. The output is a feature map in which each value corresponds to a specific patch in the input image. The final adversarial loss is computed over all patches, which encourages the generator to focus on producing realistic local details.

### 3.3. VQ-GAN Hyperparameters

We have listed the hyperparameters selected for our VQ-GAN model in Table 1.

### 3.4. The Conditional Latent Diffusion Model (cLDM)

After training the VQ-GAN model, we utilize the pre-trained latent space and apply a diffusion process to it, hence the term ‘latent diffusion’. Operating in the latent space significantly reduces computational cost. The denoising diffusion probabilistic model (DDPM) in the latent space involves two main steps: the forward (diffusion) process and the reverse (denoising) process.

In the forward process, the latent representation is gradually corrupted by the noise over a finite number of timesteps. Equation (2) illustrates this forward process.(2)$$q\left(z_{t}|z_{t-1}\right)=N\;\left(z_{t};\sqrt{\alpha_{t}}z_{t-1},\left(1-\alpha_{t}\right)I\right)$$

Here, $\alpha_{t}$ defines the noise schedule. For our experiment, we used a linear noise scheduler. The entire process is carried out over T timesteps. After the forward process, we obtain a noisy latent $z_{t}$ from the original clean latent $z_{0}$.

In the reverse process, a neural network, typically a U-Net, is trained to predict and remove the noise step-by-step, reconstructing $z_{0}$. The reverse process is represented by Equation (3).(3)$$p_{\theta }\left(z_{t-1}|z_{t},c\right)=N\;\left(z_{t-1};\mu_{\theta }\left(z_{t},t,c\right),\Sigma_{\theta }\left(z_{t},t,c\right)\right)$$

Once the latent representation $z_{0}$ is recovered, the pre-trained VQ-VAE decoder reconstructs the final image. The objective function for the U-Net is given by Equation (4).(4)$$L_{t}={\left\|\epsilon -\epsilon_{\theta }\left(z_{t},t,c\right)\right\|}^{2}$$

This objective is essentially the MSE between the actual and the predicted noise. Here, c represents the conditioning input to the U-Net. In our approach, we used the clustering results obtained from clustering the latent space as the conditioning input for the U-Net.

### 3.5. Information Maximization-Based Clustering

Information maximization-based clustering is an approach that learns discrete cluster assignments by maximizing the mutual information (MI) between the input data and the assigned clusters. This method ensures that clusters are both meaningful and well-separated by promoting high-confidence predictions while maintaining balanced cluster sizes. Specifically, mutual information is maximized by increasing the difference between the marginal entropy (which encourages well-balanced clusters) and the conditional entropy (which ensures each sample is confidently assigned to a single cluster) [51]. Self-augmented training can further enhance cluster stability by ensuring that cluster assignments remain consistent under data transformations such as rotation, translation, and scaling. This approach is particularly powerful in unsupervised learning scenarios, especially for representation learning and clustering in deep neural networks.

### 3.6. cLDM Hyperparameters

We have listed the hyperparameters selected for the cLDM model in Table 2.

Here, finetuning-1 and finetuning-2 refer to two additional training phases of our conditional latent diffusion model following the initial training. The reasons for these additional training steps will be explained in a later section.

## 4. Results

In this section, we present the various results obtained from our research.

### 4.1. VQ-GAN Reconstructions

In Figure 1, we present some original images from our test set, which were not used during the training of the VQ-GAN model. Their corresponding reconstructions closely resemble the originals, indicating that the VQ-GAN achieved high-quality reconstructions after training.

### 4.2. Cluster Validation

We trained our model using unlabeled histopathology images. To differentiate between the images, we experimented with cluster sets containing 10, 11, 12, 13, 14, 15, and 16 clusters. We then applied several internal cluster validation metrics to identify the optimal number of clusters. Internal cluster validation is necessary when ground truth information is unavailable, as it evaluates the similarity among objects within each cluster [52].

We used five different internal cluster validation metrics: the Calinski–Harabasz index, C index, Dunn index, Hartigan index, and McClain–Rao index. The Calinski–Harabasz (CH) index assesses cluster quality by measuring the ratio of between-cluster dispersion to within-cluster dispersion [53]. The CH index is defined as follows:(5)$$CH=\frac{trace(B_{k})}{trace(W_{k})}\times \frac{n-k}{k-1}$$

Here, $B_{k}$ is the between-cluster scatter matrix, which measures how well-separated the clusters are, while $W_{k}$ is the within-cluster scatter matrix, which measures the compactness of each cluster, *n* is the total number of samples, and k is the number of clusters. A higher CH index indicates better clustering—meaning clusters are well-separated and compact—whereas a lower CH index suggests poor clustering, with overlapping or dispersed clusters. The C index is another metric for evaluating cluster quality, focusing on both compactness and separation. It compares the sum of intra-cluster distances to the best and worst possible clustering scenarios [54]. The C index is defined as follows:(6)$$C=\frac{S-S_{min}}{S_{max}-S_{min}}$$

Here, S is the sum of the intra-cluster distances, that is, the sum of the distances between all points within the same cluster, $S_{min}$ is the smallest possible sum of intra-cluster distances (the ideal case), and $S_{max}$ is the largest possible sum (the worst case). Lower C index values indicate better clustering, meaning intra-cluster distances are small and clusters are compact. Conversely, higher C index values suggest poor clustering, with data points within a cluster being far apart. The Dunn index evaluates the quality of clustering by calculating the ratio of the smallest inter-cluster distance to the largest intra-cluster distance [55]. It is defined as follows:(7)$$D=\frac{{min}_{i\neq j}\;d(C_{i},C_{j})}{{max}_{k}\;d(C_{k})}$$

Here, $d(C_{i},C_{j})$ is the inter-cluster distance between clusters $C_{i}$ and $C_{j}$ (typically the minimum distance between points in different clusters), and $d(C_{k})$ is the intra-cluster distance for cluster $C_{k}$ (usually the maximum distance between points within the same cluster). A higher Dunn index indicates better clustering, meaning clusters are well-separated and compact, while a lower Dunn index suggests poor clustering, with overlapping or poorly-formed clusters. The Hartigan index is calculated using the logarithmic ratio of the between-cluster sum of squares (SSB) to the within-cluster sum of squares (SSW) [56]. It is given by Equation (8):(8)$$H=log\left(\frac{SSB}{SSW}\right)$$

A higher Hartigan index suggests well-separated clusters, while a lower index indicates less distinct clustering. Finally, the McClain–Rao (MR) index is an internal cluster validation metric used to assess the quality of a clustering configuration. It evaluates the ratio of average intra-cluster distances to average inter-cluster distances, providing insight into both the compactness and separation of clusters. The MR index is defined as follows:(9)$$MR=\frac{Average\;intra\;cluster\;distance}{Average\;inter\;cluster\;distance}$$

Lower MR index values indicate better clustering quality, as they suggest that data points within clusters are closer together (greater compactness) and clusters are well-separated from one another. Higher MR index values suggest poorer clustering quality, indicating that clusters may be overlapping or not well-defined.

From Table 3, we observe that the 14-cluster set achieves the highest CH index, Dunn index, and Hartigan index, as well as the lowest C index and MR index. Therefore, 14 is the optimal number of clusters for our dataset.

### 4.3. Clustering Results

The clustering results for the 14-cluster set are presented in Figure 2.

The results of the optimal cluster set are shown, and the clustering output from this set serves as the conditioning mechanism for generating samples and for comparison with the source images during evaluation. We consulted a clinical expert, who contributed his expertise by assigning labels to each cluster; further details are provided in the Section 5.

In this study, we used information maximization-based clustering due to its effectiveness in generating balanced and well-separated clusters in an unsupervised manner. We chose this approach because it had previously performed well with this dataset. Although clustering inherently involves dimensionality reduction and abstraction, we minimized the risk of losing critical information by optimizing the number of clusters using multiple internal validation indices, ensuring that the clusters are both meaningful and representative of the data’s diversity. Additionally, we incorporated expert feedback to interpret the clusters, providing an extra safeguard against the loss of important histopathological features.

### 4.4. Generated Image Evaluation

We generated 10,000 images after training our cLDM for each cluster set. Using the latent features of the images in the training set, we applied the clustering and then used the clustering results as conditioning inputs for generating new samples with each model. To evaluate the quality of the generated images for each cluster set, we employed three different metrics: SSIM (structural similarity index measure), MS-SSIM (multi-scale structural similarity index measure), and LPIPS (learned perceptual image patch similarity) metric. LPIPS can be used both as a model for image enhancement and as a metric for image quality assessment; previously, we used LPIPS as a model component in our VQ-GAN architecture.

SSIM evaluates image quality based on structural information rather than simple pixel-wise differences, such as those measured by MSE [57]. It takes into account luminance, contrast, and structural similarities between images, making it useful for assessing how well a generated image preserves the structure of the reference image. SSIM is particularly useful for detecting distortions. MS-SSIM is an extension of SSIM that evaluates images at multiple scales, enhancing its robustness to variations in viewing conditions [58]. By capturing details at different resolutions, MS-SSIM provides a more comprehensive evaluation of high-resolution images and complex textures. LPIPS is a deep learning-based metric that measures perceptual similarity by comparing feature representations extracted from deep neural networks. Unlike SSIM and MS-SSIM, which rely on traditional structural comparisons, LPIPS uses deep features to assess visual similarity in a manner that aligns more closely with human perception. This makes it especially valuable for evaluating images generated by modern deep learning models, such as GANs and diffusion models, where pixel-wise comparisons alone may not be sufficient.

We have listed the SSIM, MS-SSIM, and LPIPS values for each cluster set in Table 4.

Higher SSIM and MS-SSIM values (closer to 1) indicate that the generated image is structurally more similar to the reference image, reflecting better image quality. Conversely, a lower LPIPS score means the generated image is perceptually more similar to the real image. For the 14-cluster set, we achieved both the highest SSIM and MS-SSIM values, as well as the lowest LPIPS score. These results confirm that the cluster configuration with the highest quality partitioning also yields the best conditional image generation for our unlabeled dataset. Moreover, by using SSIM, MS-SSIM, and LPIPS, we ensure that both structural fidelity and perceptual realism are accounted for in our evaluations.

The authors of [59] evaluated various deep learning-based compression methods and reported that their model achieved higher MS-SSIM values and lower LPIPS scores compared to baseline methods, indicating better preservation of structural details and perceptual similarity across all compression rates. In [60], the researchers proposed StainGAN—an automated, unpaired, end-to-end stain style transfer solution inspired by CycleGAN—that preserves tissue structure without requiring a reference template. StainGAN achieved SSIM scores 10% higher than those of traditional stain normalization techniques, demonstrating a notable improvement in structural similarity to target images. Liang et al. introduced SSIM-GAN, which incorporated SSIM into its reconstruction loss to better maintain structural content [61]. Their results showed that SSIM-GAN ensures greater structural fidelity compared to baseline models. Another study addressed the need for reliable evaluation metrics in medical image synthesis [62], showing that SSIM and MS-SSIM effectively capture structural fidelity but are sensitive to preprocessing steps, whereas LPIPS better aligns with perceptual quality, particularly under realistic distortions. While none of these studies utilized clustering, our research employed clustering as a conditioning mechanism for image synthesis. This comparison highlights that our conditional latent diffusion model achieves competitive results in terms of image quality and structural fidelity, particularly given the unlabeled nature of our dataset.

### 4.5. Conditional Image Sampling Using the 14-Cluster Set

We generated 100 samples for each cluster using cluster IDs 0 to 13, as the 14-cluster set was identified as optimal in our experiment. After generating these samples for each cluster, we also assessed how many were correctly predicted by our model. Representative samples that closely resemble each cluster are shown in Figure 3.

Figure 3 displays a selection of generated samples produced using the cluster IDs. We observe that these samples exhibit similar tissue patterns to the corresponding clusters shown in Figure 2. For most clusters, the model predicted the sample IDs with high accuracy—over 70%. In several clusters, the accuracy exceeds 60%, while only one cluster shows a lower accuracy of 54%.

## 5. Discussion

As previously mentioned, we finetuned our cLDM model twice following the initial training. In this section, we explain the rationale behind these two additional finetuning steps.

Figure 4 illustrates samples generated by conditioning on the latent features of selected test images using the 14-cluster set, following the initial training, finetuning-1, and finetuning-2. The original test samples are shown in Figure 4a. As seen in Figure 4b, the quality of the generated images after the first training is quite poor. During this initial phase, our model was trained for 600 epochs; however, the MSE loss (the difference between the actual and predicted noise) remained too high, resulting in low-quality outputs. To address this issue, we froze the parameters of the neural network responsible for clustering and proceeded to finetune our model, focusing specifically on reducing the MSE loss.

We finetuned our model (finetuning-1) for 130 epochs to improve the quality of the generated samples by further reducing the MSE loss. The results after finetuning-1 are shown in Figure 4c, where the images are noticeably better than those produced after the initial training. However, the color distribution in some samples is still suboptimal and visibly inconsistent. This issue arose due to a high transformation loss from the information maximization-based clustering. To address this, we conducted an additional finetuning (finetuning-2) for 70 epochs, this time focusing on reducing the transformation loss and enhancing the color quality of the generated images. During finetuning-2, we did not freeze any model parameters. The results after this second finetuning are presented in Figure 4d.

Although our dataset lacks annotations, we consulted with a specialist to help interpret the images within each cluster shown in Figure 2. The specialist provided descriptions for each cluster, which are presented in Table 5.

These descriptive labels, based on the expert pathologist’s review, add further clarity and interpretability to the synthetic image generation process.

While our study focused on histopathology images, the proposed conditional latent diffusion model is fundamentally adaptable to other medical imaging modalities, such as radiology and dermatology. The core components, including latent space compression utilizing VQ-GAN, unsupervised clustering, and diffusion-based generation, are not specific to histopathology and can be retrained on different image types to capture modality-specific features. However, we acknowledge that each modality presents unique challenges, including variations in resolution, texture, anatomical diversity, and potential imaging artifacts, which may require adjustments to the autoencoder architecture, clustering strategies, or preprocessing steps. We anticipate that, with careful retraining, domain-specific cluster validation, and expert input for interpretation, our method could be effectively extended to other medical imaging modalities.

Finally, we highlight the main difference between our research and several recent techniques that utilize LDMs. Unconditional LDMs have demonstrated impressive image synthesis capabilities; however, they lack any conditioning mechanism, which limits their usefulness for targeted image generation tasks. Class-conditional LDMs use labels for conditioning, enabling more specific output generation, but this approach depends on annotated datasets that are often unavailable in medical imaging. Other conditioning techniques in LDMs, such as those for super-resolution, inpainting, or mask-based synthesis, have been successfully applied to enhance image details or enable region-specific synthesis using explicit inputs or guidance (e.g., text, segmentation masks, or partial images). By contrast, our method introduces clustering-based conditioning within the latent space, enabling effective and interpretable image synthesis even in the absence of annotations. This approach distinguishes our work by allowing for the generation of diverse and clinically meaningful synthetic images from unlabeled data, while still leveraging the strengths of LDMs in terms of image quality and flexibility.

## 6. Conclusions

In this work, we conditionally generated unlabeled histopathology images using our cLDM model. We began by selecting a VQ-GAN architecture as the autoencoder for our latent diffusion model, which enabled us to reconstruct very high-quality samples. Next, we performed a forward diffusion process on the latent space extracted from the pre-trained VQ-GAN, using a linear scheduler. We then applied a deep learning-based clustering technique to the latent space, and the resulting clusters were used to guide the U-Net in generating targeted outputs. Since our dataset lacked annotations, this conditioning procedure allowed us to effectively direct the generation process. During the reverse process, the U-Net learned to reconstruct a clean latent space from a noisy one, and the resulting clean latent image was passed through the pre-trained VQ-VAE decoder to generate a new image.

We experimented with different numbers of clusters and determined the optimal number using five internal cluster validation methods. All metrics consistently indicated that 14 was the optimal cluster set. Subsequently, we generated a large number of synthetic images, equal to the number of training images, within the context of each cluster set, using the latent features of the training images as conditioning inputs. The quality of these artificially generated images was evaluated using three different metrics, all of which confirmed that the images generated with the 14-cluster set were most similar to the training dataset. By incorporating expert pathologist insights into the cluster assignments, we were able to link data-driven groupings with meaningful histopathological features, thereby enhancing both the interpretability and the relevance of our synthetic image generation process.

Possible future research directions include extending our approach by incorporating semi-supervised and weakly supervised strategies, which would allow for the integration of limited annotated data alongside unlabeled samples to achieve even more refined and clinically relevant image synthesis. Additionally, our methodology can be adopted to analyze other types of medical images. Other deep learning models, beyond LDMs, could also be explored for this purpose. Finally, future work should focus on further clinical validation through collaboration with experts, in order to assess the interpretability and utility of synthetic image generation in real-world diagnostic settings.

In our study, all synthetic images were generated solely for research purposes using unlabeled data and were not intended for direct clinical decision-making. We acknowledge that deploying synthetic images in clinical settings requires strict validation to ensure their reliability and safety. Therefore, we recommend that synthetic images undergo thorough expert review and quantitative validation before any clinical application. Additionally, transparency in data provenance and clear labeling of synthetic content are essential to prevent misuse. We emphasize that our methodology should remain a research tool until comprehensive validation protocols for clinical use are established.

We have provided the architecture of our cLDM framework, the loss function used to train it, and some additional information regarding the linear scheduler in the Appendix A.

## Figures and Tables

**Figure 1 bioengineering-12-00764-f001:**
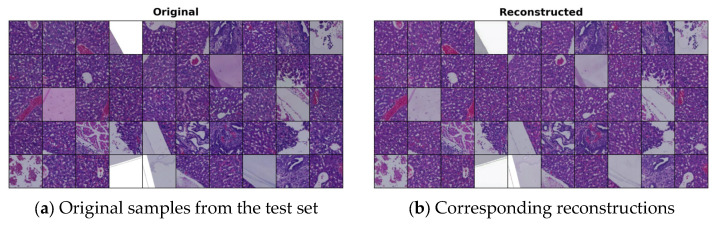
Reconstructed samples generated by the VQ-GAN model: (**a**) randomly selected samples from the test set; and (**b**) corresponding reconstructions by VQ-GAN.

**Figure 2 bioengineering-12-00764-f002:**
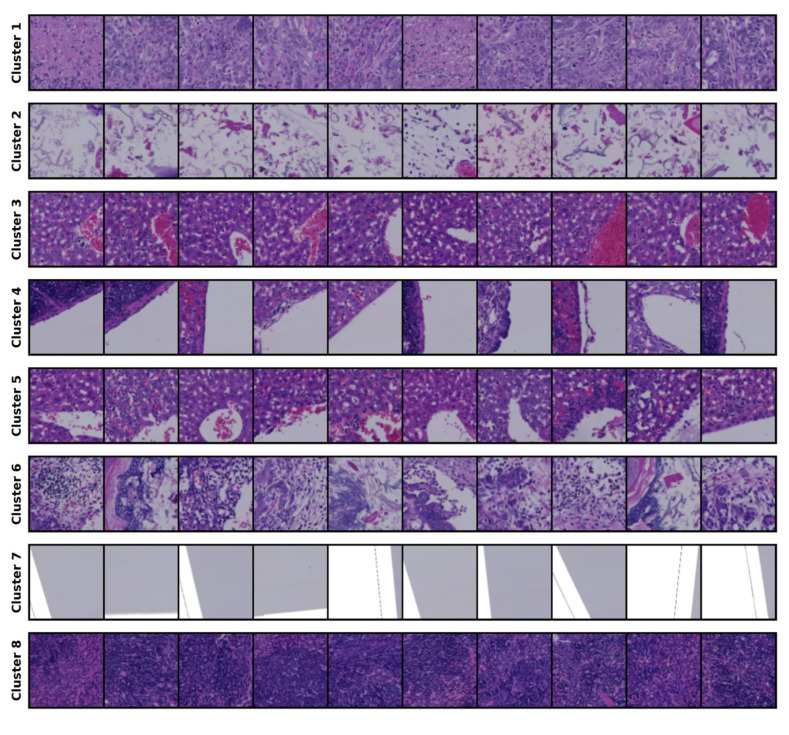
Clustering results for the 14-cluster set.

**Figure 3 bioengineering-12-00764-f003:**
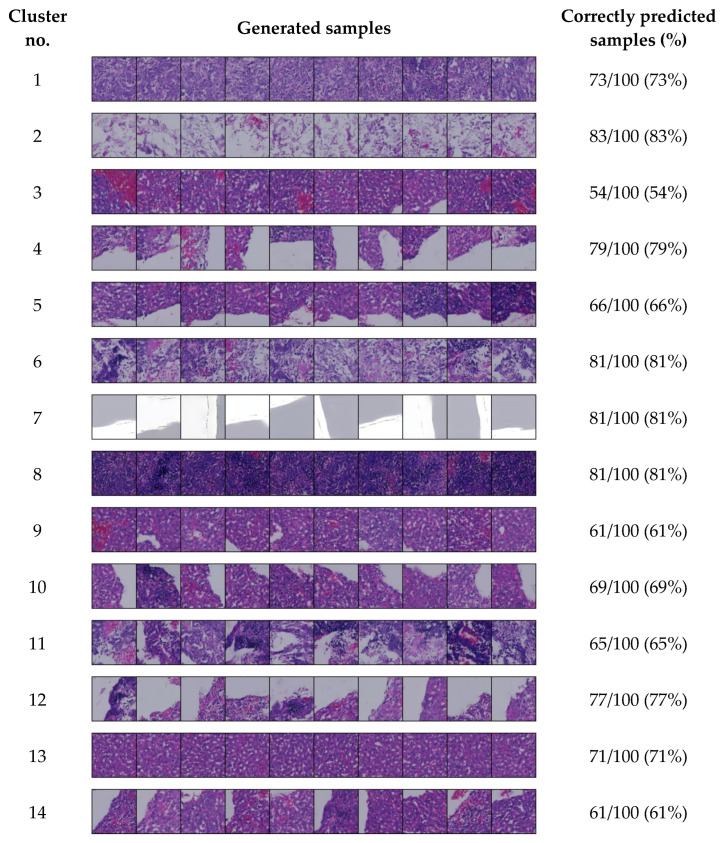
Conditionally generated samples for each cluster in the 14-cluster set.

**Figure 4 bioengineering-12-00764-f004:**
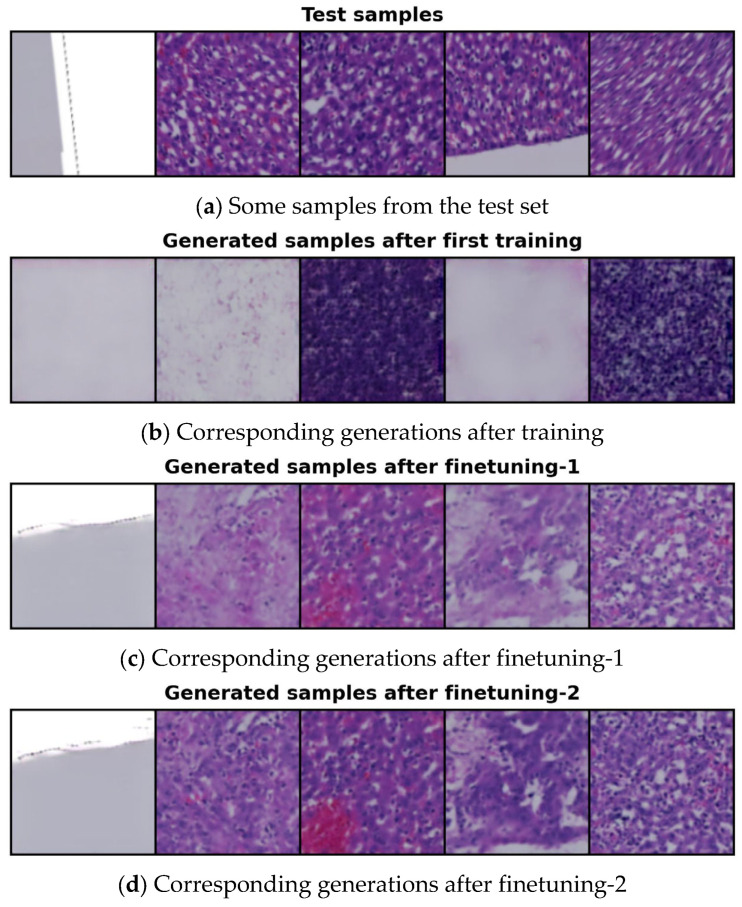
Comparison of conditionally generated samples: (**a**) random samples from the test set; (**b**) samples generated after initial training; (**c**) samples generated after finetuning-1; and (**d**) samples generated after finetuning-2.

**Table 1 bioengineering-12-00764-t001:** Hyperparameters for the VQ-GAN.

Hyperparameter	Value
Codebook size	128
Codebook vector dimension	16
Latent space resolution	32 × 32
Batch size	16
Learning rate	0.0001
Epochs	20
Optimizer	Adam
Activation function	SiLU

**Table 2 bioengineering-12-00764-t002:** Hyperparameters for cLDM.

Hyperparameter	Value
Batch size	16
Activation function	SiLU
Timesteps	1000
Noise scheduler	Linear
U-Net bottleneck resolution	8 × 8
Learning rate (cLDM training)	0.003
Optimizer (cLDM training)	Adadelta
Epochs (cLDM training)	600
Learning rate (cLDM finetuning-1)	0.0001
Optimizer (cLDM finetuing-1)	Adam
Epochs (cLDM finetuning-1)	130
Learning rate (cLDM finetuning-2)	0.003
Optimizer (cLDM finetuing-2)	Adadelta
Epochs (cLDM finetuning-2)	70

**Table 3 bioengineering-12-00764-t003:** Internal validation metrics for different cluster sets.

No. of Clusters	CH Index	C Index	Dunn Index	Hartigan Index	MR Index
10	2142.54026	0.12234	0.00441	0.65763	0.54631
11	2866.86433	0.10477	0.00477	1.05432	0.49791
12	1865.81774	0.12042	0.00463	0.72021	0.55443
13	2836.22720	0.09131	0.00424	1.22610	0.46335
**14**	**3367.74211**	**0.07645**	**0.00548**	**1.47801**	**0.41643**
15	2027.82944	0.10489	0.00424	1.04494	0.51414
16	2603.28507	0.08615	0.00490	1.36384	0.45223

We have summarized the values of the five internal cluster validation metrics for all cluster sets in Table 3.

**Table 4 bioengineering-12-00764-t004:** Generated image evaluation metrics for different cluster sets.

No. of Clusters	SSIM	MS-SSIM	LPIPS
10	0.7814	0.2138	0.5008
11	0.7835	0.2130	0.5003
12	0.7801	0.2168	0.5032
13	0.7865	0.2161	0.4987
**14**	**0.7988**	**0.2196**	**0.4923**
15	0.7889	0.2152	0.4980
16	0.7908	0.2186	0.4939

**Table 5 bioengineering-12-00764-t005:** Descriptions for clusters in the 14-cluster set.

Cluster No.	Description from Expert
1	Tissue with high cancer cell density
2	Tissue with low cell density
3	Liver parenchyma with red blood cells
4	Tissue with large voids
5	Tumor area with voids
6	Tumor with abundant stromal components
7	Surrounding empty space
8	Tissue with high cell density
9	Tissue containing small voids
10	Boundary of tissue containing red blood cells
11	Voids within the tissue
12	Boundary between tissue and empty space
13	Normal liver parenchyma
14	Edge of the liver

Cluster descriptions provided by the expert are presented in Table 5.

## Data Availability

If this document is accepted, we will provide our dataset that can be accessed from public domain resources. In addition, the code will be made public and can be obtained from GitHub, https://github.com/randomaccess2023/KPC_LDM_128, accessed on 13 July 2025.

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
