# Peer review of "Implementation of a Conditional Latent Diffusion-Based Generative Model to Synthetically Create Unlabeled Histopathological Images"

_bioengineering, 2025, doi:10.3390/bioengineering12070764_

Round 1

Reviewer 1 Report

Comments and Suggestions for Authors
  • What is the main question addressed by the research?

The publication concerns the use of generative artificial intelligence and the use of deep learning techniques for the analysis of graphical data. The authors developed a model that allows the analysis of histopathological images. The VQ-GAN mechanism was used and then fusion in the latent space was performed. The article presents how - using training of the proposed model to generate a set of clusters and automate the classification of histopathological images.

  • Do you consider the topic original or relevant to the field? Does it address a specific gap in the field? Please also explain why this is/ is not the case. • What does it add to the subject area compared with other published material?

Very rapidly developing diffusion models were used to improve the fidelity of the acquired images. The authors used a latent space diffusion model using Structural Simliarity Index Measure and Learned Perceptual Image Path Similarity, which gave satisfactory results of generated images while ensuring good structural fidelity. The authors present a step-by-step method of getting from data to the result, which is particularly valuable because it can be practically used by other researchers.

  • What specific improvements should the authors consider regarding the methodology? What further controls should be considered?

The method of analysis and methodology provided by the authors is sufficient and adequately described and there is no need to extend it.

At the end of the publication, the authors include a list of abbreviations used, which greatly facilitates the analysis of the text.

  • Are the conclusions consistent with the evidence and arguments presented and do they address the main question posed? Please also explain why this is/is not the case.

The indicators illustrating the obtained results are given in tabular form. The obtained results are also illustrated in graphical and descriptive form. The obtained results have been analyzed. Appropriate conclusions have been provided. The architecture of the diffusion model is provided as additional materials.

The authors' declaration regarding making the data set and the developed code available in the public domain is very valuable.

  • Are the references appropriate?

The list of references includes 46 items. The vast majority of them are articles from the last few years.

Most of the references (thirty-one items) are cited in the first chapter, where the authors discuss research conducted in other research centers.

The authors use the next references to justify the proposed computational model.

Item [32] is a self-quotation of the authors of this article. This can be considered an acceptable practice.

Final conclusion:

The article is scientifically correct. The article is also correct from an editorial perspective. The whole thing meets the criteria required for scientific works and in my opinion qualifies for publication.

Author Response

Comment 1: The publication concerns the use of generative artificial intelligence and the use of deep learning techniques for the analysis of graphical data. The authors developed a model that allows the analysis of histopathological images. The VQ-GAN mechanism was used and then fusion in the latent space was performed. The article presents how - using training of the proposed model to generate a set of clusters and automate the classification of histopathological images.

Response 1: We thank the reviewer for accurately summarizing our work. We would like to clarify that, although our approach uses clustering of latent features as a conditioning mechanism for generating synthetic histopathology images, the primary goal is not automated classification. Rather, our main objective is to enable structure-aware conditional generation in the absence of annotated data.

Comment 2: Very rapidly developing diffusion models were used to improve the fidelity of the acquired images. The authors used a latent space diffusion model using Structural Simliarity Index Measure and Learned Perceptual Image Path Similarity, which gave satisfactory results of generated images while ensuring good structural fidelity. The authors present a step-by-step method of getting from data to the result, which is particularly valuable because it can be practically used by other researchers.

Response 2: We sincerely thank the reviewer for the positive and encouraging feedback regarding the originality and significance of our work. We appreciate the recognition of our step-by-step methodology and practical approach as valuable contributions to the research community. We are also pleased that our use of latent space diffusion models and relevant evaluation metrics was acknowledged as effectively addressing a key gap in the field.

Comment 3: The method of analysis and methodology provided by the authors is sufficient and adequately described and there is no need to extend it.

At the end of the publication, the authors include a list of abbreviations used, which greatly facilitates the analysis of the text.

Response 3: We thank the reviewer for their positive assessment of our methodology and for acknowledging the usefulness of the included list of abbreviations. We are pleased that our methodological descriptions were found to be sufficient and clear.

Comment 4: The indicators illustrating the obtained results are given in tabular form. The obtained results are also illustrated in graphical and descriptive form. The obtained results have been analyzed. Appropriate conclusions have been provided. The architecture of the diffusion model is provided as additional materials.

The authors' declaration regarding making the data set and the developed code available in the public domain is very valuable.

Response 4: We thank the reviewer for their positive assessment of our results, analysis, and conclusions. We appreciate their recognition of our efforts to present the data and findings clearly, as well as their acknowledgment of our commitment to open science by making our dataset and developed code publicly available.

Comment 5: The list of references includes 46 items. The vast majority of them are articles from the last few years.

Most of the references (thirty-one items) are cited in the first chapter, where the authors discuss research conducted in other research centers.

The authors use the next references to justify the proposed computational model.

Item [32] is a self-quotation of the authors of this article. This can be considered an acceptable practice.

Response 5: We appreciate the reviewer's careful assessment of our references and are pleased that the selected literature and citations were found to be relevant and appropriate to our work. We also acknowledge the reviewer’s remark regarding our self-citation and thank them for considering it acceptable in this context.

Comment 6: The article is scientifically correct. The article is also correct from an editorial perspective. The whole thing meets the criteria required for scientific works and in my opinion qualifies for publication.

Response 6: We sincerely thank the reviewer for their positive final evaluation. We are pleased that our work is considered both scientifically and editorially sound, and we greatly appreciate the recommendation for publication.

Reviewer 2 Report

Comments and Suggestions for Authors

Rumman et al. analyzed in the proposed paper generative image models using unlabeled histopathology images and determined the optimal number of clusters.

Parts of the paper should be improved and restructured.

Punctually, these are:

1. More clearly for the reader is to split a large introduction into two sections. Introduction and Related Work, the section Related work has to contain the state-of-the-art papers with their strengths and weaknesses.

  1. The introduction section should contain the information about vulnerabilities of GAI algorithms. A large description can be found in the papers https://doi.org/10.3390/bdcc9020045 and https://doi.org/10.3390/electronics14071270.
  2. The last part of the introduction section should contain the main contributions and also the proposed scopes.
  3. Besides the proposed, please add two other GAI algorithms for comparing the performance in terms of the same metrics for clusters (Calinski-Harabasz index, C index, Dunn index, Hartigan index, and Mclain-Rao) and similarity between images (SSIM (Structural Similarity Index Measure), MS-SSIM (Multi-Scale Structural Similarity Index Measure), and LPIPS (Learned Perceptual Image Patch Similarity)).
  4. Please explain why in figure "Figure 2. Clustering results of the 14-cluster set," cluster 7 does not contain histopathological structure. Also, in „Figure 3. Conditionally generated samples based on the 14-cluster set,” the same problem with cluster 7.
  5. In the section „3.4. Generated Image Evaluation,” the SSIM index is not referenced; it is described in detail in the paper https://doi.org/10.3390/jimaging10090235.
  6. In section „4. Discussion,” please compare your study with methods from scientific literature.
  7. Please mention the future research that will have a connection with your proposal.

Author Response

Comment 1: More clearly for the reader is to split a large introduction into two sections. Introduction and Related Work, the section Related work has to contain the state-of-the-art papers with their strengths and weaknesses.

Response 1: We thank the reviewer for this constructive suggestion. In response, we have revised the manuscript by dividing the original Introduction into two distinct sections: "Introduction" and "Related Work." The "Related Work" now presents an overview of recent state-of-the-art studies in the field, including a brief discussion of their respective strengths and weaknesses.

Comment 2: The introduction section should contain the information about vulnerabilities of GAI algorithms. A large description can be found in the papers https://doi.org/10.3390/bdcc9020045 and https://doi.org/10.3390/electronics14071270.

Response 2: We appreciate the reviewer's suggestion to include information about the vulnerabilities of GAI algorithms in the "Introduction" section. In accordance with this recommendation, we have added two new paragraphs that highlight key issues, such as feature corruption leading to degradation in classifier performance and vulnerabilities associated with the use of synthetic medical data. We have also cited the recommended references to provide further context.

Comment 3: The last part of the introduction section should contain the main contributions and also the proposed scopes.

Response 3: We have revised the "Introduction" section in accordance with the reviewer's suggestion and now included the main contributions of our research at the end of the "Introduction" section. The scope of this study focuses on synthetic image generation for histopathological data without the need for manual annotations, with potential applicability to other domains where labeled data is limited. We also clarify that our primary emphasis is on image generation rather than automated diagnosis.

Comment 4: Besides the proposed, please add two other GAI algorithms for comparing the performance in terms of the same metrics for clusters (Calinski-Harabasz index, C index, Dunn index, Hartigan index, and Mclain-Rao) and similarity between images (SSIM (Structural Similarity Index Measure), MS-SSIM (Multi-Scale Structural Similarity Index Measure), and LPIPS (Learned Perceptual Image Patch Similarity)).

Response 4: Thank you very much for this valuable and insightful suggestion. We fully agree that a comparative evaluation with other GAI models, such as GAN and VAE, would provide important additional perspectives on the performance of our approach. Ideally, we would have liked to include such comparisons. However, due to the complexity of integrating information maximization-based clustering mechanisms into these alternative models, and the considerable time required for proper hyperparameter tuning and validation, especially since optimizing the current cLDM framework already demanded several months of dedicated work, we regret that a thorough and fair implementation of additional GAI models is not feasible within the timeframe of this revision. Recognizing the importance of enriching our method's evaluation, we carried out additional analyses using alternative clustering algorithms within our current framework. Specifically, we utilized K-Means and Gaussian Mixture Model (GMM) clustering approaches, evaluating the clustering results using multiple internal validation metrics (Calinski-Harabasz index, C index, Dunn index, Hartigan index, and Mclain-Rao index). However, the metrics didn't unanimously agree on the optimal number of clusters. For K-Means, the Calinski-Harabasz index selected 10 as the optimal number of clusters, while the other metrics identified 16. For GMM clustering, the Calinski-Harabasz index and Dunn index indicated 10 and 15 clusters as optimal, respectively, while the remaining three metrics preferred 16 clusters. K-Means and GMM are less sophisticated algorithms compared to our information maximization-based clustering procedure, and they struggle to distinguish the complex features present in our dataset as effectively as our chosen method. These additional analyses provide further insight into the stability and robustness of our clustering strategy. Notably, using only 10 clusters resulted in a lack of diversity and frequent mixing of different tissue types within single clusters, which diminished interpretability and biological relevance. Conversely, increasing the number of clusters to 16 led to repetitive clusters with limited added value. We sincerely appreciate the reviewer's thoughtful recommendation on this current limitation (the implementation of other GAI models for comparison). We plan to explore these comparative studies in our future work.

Comment 5: Please explain why in figure "Figure 2. Clustering results of the 14-cluster set," cluster 7 does not contain histopathological structure. Also, in "Figure 3. Conditionally generated samples based on the 14-cluster set," the same problem with cluster 7.

Response 5: We thank the reviewer for highlighting this important observation regarding Cluster 7. The dataset used in this study was randomly created from whole-slide images (WSIs). Although we removed most of the empty spaces extracted from these WSIs during dataset preparation, some non-tissue regions remained due to the random sampling process. Cluster 7 specifically captures these non-tissue or empty-space areas, which do not contain histopathological structures. Having such a cluster is beneficial for patch-based WSI analysis, which we have explored in our previous research. Consequently, the conditionally generated samples for Cluster 7 naturally reflect these non-tissue characteristics.

Comment 6: In the section "3.4. Generated Image Evaluation," the SSIM index is not referenced; it is described in detail in the paper https://doi.org/10.3390/jimaging10090235.

Response 6: Thank you for pointing out the need to reference the SSIM index in Subsection "Generated Image Evaluation". After reviewing the suggested paper (https://doi.org/10.3390/jimaging10090235), we found that the original SSIM metric is more accurately described in the foundational work by Wang et al. (IEEE Transactions on Image Processing, 2004), which we have already cited in our manuscript in the same Subsection when discussing the SSIM index. If preferred, we can also include the suggested paper (https://doi.org/10.3390/jimaging10090235) as an additional reference in the "Introduction" section.

Comment 7: In section "4. Discussion," please compare your study with methods from scientific literature.

Response 7: We thank the reviewer for this valuable suggestion. In response, we have expanded the "Discussion" section to include a comparison of our proposed approach with several relevant methods from recent literature that utilize LDMs.

Comment 8: Please mention the future research that will have a connection with your proposal.

Response 8: We sincerely thank the reviewer for this suggestion. We have added a statement in the "Conclusion" section outlining possible future research directions that could serve as extensions of our approach.

Reviewer 3 Report

Comments and Suggestions for Authors

The research concentrates on histopathology images, which possess distinct characteristics.  The applicability of the proposed conditional latent diffusion model to other categories of medical images, including radiological or dermatological images, is ambiguous.  In what manner does the authors' methodology address the diversity and variability of alternative imaging modalities in healthcare?  What is the authors' strategy for extending the conditional latent diffusion model to additional healthcare imaging modalities?  Do they anticipate any obstacles in extending their methodology beyond histopathology?

 What is the robustness of the clustering methodology employed in the study?  What is the performance of the authors' model when applied to various clustering algorithms?  Is there a potential risk of losing critical information due to clustering, and how do the authors address this concern?

 In what manner do the authors tackle the ethical implications of employing synthetic images in healthcare environments?  What protocols are implemented to guarantee that synthetic images serve as dependable references in clinical applications?

 How do the authors guarantee that the synthetic images produced by the model are both realistic and applicable in a clinical setting, given the absence of labeled data?  What protocols exist to ensure the interpretability of these generated images, particularly when expert labeling is not utilized during training?

 Could the authors provide further details regarding the specific quantitative evaluation methods employed to assess the quality of the synthetic images?  How do they guarantee the clinical relevance of these methods and ensure that the produced images comply with healthcare standards?

Author Response

Comment 1: The applicability of the proposed conditional latent diffusion model to other categories of medical images, including radiological or dermatological images, is ambiguous. In what manner does the authors' methodology address the diversity and variability of alternative imaging modalities in healthcare? What is the authors' strategy for extending the conditional latent diffusion model to additional healthcare imaging modalities? Do they anticipate any obstacles in extending their methodology beyond histopathology?

Response 1: We thank the reviewer for this thoughtful question. While our study focused on histopathology images, the proposed conditional latent diffusion model is fundamentally adaptable to other medical imaging modalities, such as radiology and dermatology. The core components: latent space compression using VQ-GAN, unsupervised clustering, and diffusion-based generation, are not specific to histopathology and can be retrained on alternative image types to capture modality-specific features. However, we acknowledge that each modality presents unique challenges, including differences in resolution, texture, anatomical diversity, and potential imaging artifacts, which may require adjustments to the autoencoder architecture, clustering strategies, or pre-processing steps. We anticipate that careful retraining, domain-specific cluster validation, and expert input for interpretation will be necessary for effectively extending our method to new modalities. We have added further information regarding this point in the "Discussion" section of the revised manuscript to highlight both the model's adaptability and the potential obstacles involved in generalizing to other types of healthcare imaging data.

Comment 2: What is the robustness of the clustering methodology employed in the study? What is the performance of the authors' model when applied to various clustering algorithms? Is there a potential risk of losing critical information due to clustering, and how do the authors address this concern?

Response 2: We appreciate the reviewer's question regarding the robustness of our clustering methodology. In this study, we employed information maximization-based clustering because of its effectiveness in generating balanced and well-separated clusters in an unsupervised setting. We selected this approach based on its strong performance in our prior work with this dataset. While clustering inherently involves some degree of dimensionality reduction and abstraction, we addressed the potential risk of losing critical information by carefully optimizing the number of clusters using multiple internal validation indices. This approach ensures that the clusters are both meaningful and representative of the data's diversity. Additionally, we incorporated expert feedback to interpret the resulting clusters, providing further assurance that essential histopathological features are preserved. We have included these details in Subsection "Clustering Results" of the revised manuscript. We also evaluated two other clustering algorithms: K-means and Gaussian Mixture Model (GMM) clustering. However, the internal validation metrics didn't unanimously agree on the optimal number of clusters for these methods, whereas our information maximization-based approach yielded consistent results across the metrics.

Comment 3: In what manner do the authors tackle the ethical implications of employing synthetic images in healthcare environments? What protocols are implemented to guarantee that synthetic images serve as dependable references in clinical applications?

Response 3: We thank the reviewer for highlighting the important ethical considerations associated with the use of synthetic images in healthcare. In our study, all synthetic images were generated solely for research purposes using unlabeled data and were not intended for direct clinical decision-making. We recognize that deploying synthetic images in clinical settings requires strict validation to ensure their reliability and safety. Therefore, we advocate that synthetic images should undergo rigorous expert review and comprehensive quantitative validation before being considered for any clinical application. Additionally, maintaining transparency regarding data provenance and clearly labeling synthetic content are essential to prevent misuse. We have included a statement in the revised "Conclusion" section to address these ethical considerations and to emphasize that our methodology should remain a research tool until robust validation protocols for clinical use are established.

Comment 4: How do the authors guarantee that the synthetic images produced by the model are both realistic and applicable in a clinical setting, given the absence of labeled data? What protocols exist to ensure the interpretability of these generated images, particularly when expert labeling is not utilized during training?

Response 4: We thank the reviewer for this valuable question. In the absence of labeled data, we ensured the realism of the synthetic images by using multiple quantitative evaluation metrics (SSIM, MS-SSIM, and LPIPS) to assess both structural fidelity and perceptual similarity to real images. To further support clinical applicability and interpretability, we incorporated expert pathologist review after clustering, enabling us to associate cluster outputs with meaningful histopathological features. Although expert labels were not used during model training, their input provided an additional layer of validation and interpretability. We acknowledge that direct clinical deployment would require further validation and ongoing expert involvement.

Comment 5: Could the authors provide further details regarding the specific quantitative evaluation methods employed to assess the quality of the synthetic images? How do they guarantee the clinical relevance of these methods and ensure that the produced images comply with healthcare standards?

Response 5: We appreciate the reviewer's request for further clarification regarding our evaluation methods. In our study, we assessed the quality of synthetic images using three widely recognized quantitative metrics: SSIM, MS-SSIM, and LPIPS. These metrics were selected because they evaluate both the structural integrity and perceptual similarity between generated and real images—attributes that are crucial for visual fidelity in medical imaging. While these quantitative measures provide objective assessments, we acknowledge that clinical relevance also depends on expert interpretation and alignment with real diagnostic tasks. Therefore, in addition to quantitative evaluation, we consulted with a clinical expert to interpret and validate cluster outputs, ensuring that our synthetic images are not only visually realistic but also meaningful from a clinical standpoint. We recognize that clinical implementation would require more extensive validation and corroboration.

***I have revised the manuscript to improve the English and more clearly convey the research presented.***

Reviewer 4 Report

Comments and Suggestions for Authors

The authors present a topic of current interest Model to Synthetically Create Unlabeled Histopathological Images. The topic is topical and the article is well structured from a theoretical, practical and results presented point of view, including in the supplementary material. Since the topic is very well documented in the specialized literature, I recommend the authors to make a comparison between the results obtained by the authors with similar results reported in the specialized literature. From this comparison it should result - why the presented results are better than other similar results from the specialized literature.

Author Response

Comment: The authors present a topic of current interest Model to Synthetically Create Unlabeled Histopathological Images. The topic is topical and the article is well structured from a theoretical, practical and results presented point of view, including in the supplementary material. Since the topic is very well documented in the specialized literature, I recommend the authors to make a comparison between the results obtained by the authors with similar results reported in the specialized literature. From this comparison it should result - why the presented results are better than other similar results from the specialized literature.

Response: We thank the reviewer for this helpful suggestion. We agree that comparing our results with those reported in recent literature provides valuable context and further highlights the significance of our approach. In response, we have added additional information to the revised Subsection "Generative Image Evaluation", citing recent studies that have used similar image evaluation metrics such as SSIM, MS-SSIM, and LPIPS for synthetic histopathology and medical images. We have also clarified the methodological differences between our research and these works.

Reviewer 5 Report

Comments and Suggestions for Authors

This manuscript presents a novel conditional latent diffusion model applied to the generation of synthetic histopathology images without labeled data. The authors combine VQ-GAN encoding with information maximization-based clustering to condition diffusion in latent space. The approach is technically sound, addresses a significant gap in unlabeled data generation, and includes expert validation, making it highly relevant to biomedical image analysis. Overall, the topic of this work is interesting, and the manuscript is well structured. The detailed comments are given as follows.

  1. What was the specific architecture of the encoder/decoder in the VQ-GAN? Were any pre-trained weights used, or was the model trained from scratch?
  2. How was the 32×32 latent space size determined? Did the authors test the impact of varying this resolution on generation quality?
  3. Was a specific clustering algorithm used for mutual information maximization? If custom, can it generalize to other domains?
  4. How sensitive are the results to changes in key hyperparameters, such as number of diffusion timesteps, learning rates, or β in the VQ-GAN loss?
  5. Why was a linear noise scheduler selected? Were other schedules like cosine or quadratic tested for comparative performance?
  6. Why was Adadelta chosen for the cLDM training? Have other optimizers like AdamW or SGD been tested?
  7. Broaden and update literature review on deep learning for image processing. 
  8. Why was the range of 10 to 16 clusters chosen for evaluation? Would a wider or adaptive search yield better segmentation?
  9. More future research should be included in conclusion part.

Author Response

Comment 1: What was the specific architecture of the encoder/decoder in the VQ-GAN? Were any pre-trained weights used, or was the model trained from scratch?

Response 1: We thank the reviewer for their question regarding the specific architecture of the encoder and decoder in the VQ-GAN model. In our study, we used the VQ-GAN architecture available from a widely recognized open-source repository, which closely follows the original VQ-GAN design and has been widely adopted in the community. The encoder consists of several convolutional layers followed by downsampling blocks to compress high-dimensional histopathology images into a lower-dimensional latent representation, while the decoder mirrors this process using upsampling layers and convolutional blocks to reconstruct images from the latent codes. Both the encoder and decoder incorporate residual connections to facilitate stable training and improve generative quality. We did not use any pre-trained weights; all components of the VQ-GAN were trained from scratch on our histopathology dataset, allowing the model to learn features specifically relevant to our domain.

Comment 2: How was the 32×32 latent space size determined? Did the authors test the impact of varying this resolution on generation quality?

Response 2: Thank you for raising this important point regarding our choice of the 32×32 latent space resolution in the VQ-GAN model. We selected the 32×32 configuration to achieve an optimal balance between compression efficiency and preservation of essential histopathological details. The resolution effectively reduces computational requirements while maintaining sufficient spatial information for high-quality image reconstruction. We also experimented with a 16×16 latent space, which produced good reconstructions; however, we observed slight visual differences compared to the results obtained with the 32×32 configuration. In other words, the 32×32 latent space yielded reconstructions that were much more similar to the original images. Using a higher latent resolution (such as 64×64) would have unnecessarily increased the computational cost for our dataset of 128×128 images, without providing significant benefits. Based on these observations, we found that the 32×32 configuration offered the best trade-off for our dataset.

Comment 3: Was a specific clustering algorithm used for mutual information maximization? If custom, can it generalize to other domains?

Response 3: We appreciate the reviewer's insightful question regarding the clustering method used for mutual information maximization. In our work, we implemented an information maximization-based clustering approach, which seeks to maximize the mutual information between input data and cluster assignments. Unlike conventional clustering algorithms such as K-Means, this method employs a neural network that learns cluster assignments by optimizing a mutual information objective. The main advantage of this approach is its ability to discover balanced and well-separated clusters, making it particularly effective for unsupervised scenarios and complex datasets like ours. While we tailored aspects of the training and loss calculation to better suit our histopathology data, the underlying method is not domain-specific and can be generalized to other image types or even non-image data, as long as the features are amenable to clustering.

Comment 4: How sensitive are the results to changes in key hyperparameters, such as number of diffusion timesteps, learning rates, or β in the VQ-GAN loss?

Response 4: We thank the reviewer for raising this question regarding the sensitivity of our results to key hyperparameters such as the number of diffusion timesteps, learning rates, and the β parameter in the VQ-GAN loss. In designing our experiments, we adopted hyperparameter settings that are widely used in established open-source repositories, particularly those with broad community adoption and demonstrated performance. For example, we set the number of diffusion timesteps to 1000 and used a learning rate of 0.0001 for VQ-GAN, both of which are standard in the literature and community implementations. These values consistently produced high-quality results on our dataset without causing model instability. The learning rates performed well without the need for extensive manual tuning. For the β parameter in the VQ-GAN loss, usually a value between 0 and 1 can be chosen. We chose a value of 0.2, which is also commonly used in open-source implementations, and resulted in stable model training. While some degree of sensitivity to hyperparameter changes is typical in deep learning models, our chosen settings proved effective and did not require additional, domain-specific optimization for our application.

Comment 5: Why was a linear noise scheduler selected? Were other schedules like cosine or quadratic tested for comparative performance?

Response 5: Thank you for raising the question regarding our choice of a linear noise scheduler in the diffusion process. We selected the linear scheduler due to its widespread use in foundational diffusion model studies and our own prior experience with the dataset. In one of our recent conference papers (not yet publicly available), we systematically compared linear, cosine, and sigmoid noise schedulers using the same histopathology image dataset and a standard DDPM model. Our results showed that the linear scheduler consistently produced superior outcomes in terms of both image fidelity and diversity. These findings informed our decision to use the linear scheduler in the present study. Nevertheless, we acknowledge that alternative schedulers such as cosine or quadratic may offer advantages in different contexts and could be explored in future work.

Comment 6: Why was Adadelta chosen for the cLDM training? Have other optimizers like AdamW or SGD been tested?

Response 6: We appreciate the reviewer's inquiry regarding our choice of Adadelta as the optimizer for cLDM training. We experimented with other commonly used optimizers, such as Adam, AdamW, and SGD. However, we found that these alternatives were not effective in maximizing the difference between marginal and conditional entropy—a key requirement for achieving well-separated and balanced clusters in our information maximization-based clustering framework. In contrast, Adadelta consistently enabled our model to optimize this objective more reliably, resulting in better cluster assignments and improved overall model performance. We believe the adaptive learning rate mechanism of Adadelta contributed to its effectiveness by promoting more stable and robust optimization. Based on these empirical results, we selected Adadelta for the main cLDM training, while Adam was still used for certain fine-tuning phases as described in the manuscript.

Comment 7: Broaden and update literature review on deep learning for image processing.

Response 7: Thank you for highlighting the need to broaden and update our literature review on deep learning for image processing. We recognize that the field—especially in generative modeling, unsupervised learning, and medical imaging—has evolved rapidly in recent years. In response to your comment, we have revised the "Related Works" section to include additional recent studies beyond those focused on diffusion models in the last three paragraphs. These updates provide a more comprehensive overview of advances in deep learning for image processing. We appreciate the reviewer's valuable suggestion and believe these additions further strengthen the manuscript.

Comment 8: Why was the range of 10 to 16 clusters chosen for evaluation? Would a wider or adaptive search yield better segmentation?

Response 8: We thank the reviewer for this thoughtful question. In our experiments, we observed that using only 10 clusters resulted in limited variety and frequent mixing of different tissue types within single clusters, reducing interpretability and biological relevance. On the other hand, increasing the number of clusters to 16 led to the emergence of repetitive clusters, which provided little additional value. The range of 10 to 16 clusters was chosen as it offered a practical balance, enabling us to capture the main morphological and histological variations present in the dataset without introducing excessive redundancy or noise. This range produced stable, interpretable, and expert-validated clusters in our application, as confirmed by internal cluster validation metrics.

Comment 9: More future research should be included in conclusion part.

Response 9: We thank the reviewer for this suggestion. In response, we have added a statement in the "Conclusion" section outlining possible future research directions that could extend our approach.

Round 2

Reviewer 2 Report

Comments and Suggestions for Authors

The paper was significantly improved. 

Reviewer 3 Report

Comments and Suggestions for Authors

The manuscript can be accepted for publication

Comments on the Quality of English Language

The English could be improved to more clearly express the research.

Reviewer 5 Report

Comments and Suggestions for Authors

The authors addressed my comments.